# Prognostic Value of Cortisol Index of Endobiogeny in Acute Myocardial Infarction Patients

**DOI:** 10.3390/medicina57060602

**Published:** 2021-06-11

**Authors:** Rima Braukyliene, Kamyar Hedayat, Laura Zajanckauskiene, Martynas Jurenas, Ramunas Unikas, Ali Aldujeli, Osvaldas Petrokas, Vytautas Zabiela, Rasa Steponaviciute, Astra Vitkauskiene, Brigita Hedayat, Sandrita Simonyte, Vaiva Lesauskaite, Jean Claude Lapraz, Diana Zaliaduonyte

**Affiliations:** 1Cardiology Department, Lithuanian University of Health Sciences, LT 50161 Kaunas, Lithuania; laurosurbo@gmail.com (L.Z.); martynas.jurenas@kaunoklinikos.lt (M.J.); ramunas.unikas@kaunoklinikos.lt (R.U.); ali.aldujeli@kaunoklinikos.lt (A.A.); osvaldas.petrokas@gmail.com (O.P.); vytautas.zabiela@kaunoklinikos.lt (V.Z.); rasa.steponaviciute@kaunoklinikos.lt (R.S.); astra.vitkauskiene@kaunoklinikos.lt (A.V.); Sandrita.Simonyte@lsmuni.lt (S.S.); vaiva.lesauskaite@lsmuni.lt (V.L.); diana.zaliaduonyte@kaunoklinikos.lt (D.Z.); 2Systems Biology Research Group, Chicago, IL 60603, USA; kmhedayat@fshcenter.com (K.H.); brigita@fshcenter.com (B.H.); jeanclaudlapraz@icloud.com (J.C.L.)

**Keywords:** acute myocardial infarction, serum cortisol, cortisol index

## Abstract

*Background and Objectives*: Serum cortisol has been extensively studied for its role during acute myocardial infarction (AMI). Reports have been inconsistent, with high and low serum cortisol associated with various clinical outcomes. Several publications claim to have developed methods to evaluate cortisol activity by using elements of complete blood count with its differential. This study aims to compare the prognostic value of the cortisol index of Endobiogeny with serum cortisol in AMI patients, and to identify if the risk of mortality in AMI patients can be more precisely assessed by using both troponin I and cortisol index than troponin I alone. *Materials and methods*: This prospective study included 123 consecutive patients diagnosed with AMI. Diagnostic coronary angiography and revascularization was performed for all patients. Cortisol index was measured on admission, on discharge, and after 6 months. Two year follow-up for all patients was obtained. *Results*: Our study shows cortisol index peaks at 7–12 h after the onset of AMI, while serum cortisol peaked within 3 h from the onset of AMI. The cortisol index is elevated at admission, then significantly decreases at discharge; furthermore, the decline to its bottom most at 6 months is observed with mean values being constantly elevated. The cortisol index on admission correlated with 24-month mortality. We established combined cut-off values of cortisol index on admission > 100 and troponin I > 1.56 μg/las a prognosticator of poor outcomes for the 24-month period. *Conclusions*: The cortisol index derived from the global living systems theory of Endobiogeny is more predictive of mortality than serum cortisol. Moreover, a combined assessment of cortisol index and Troponin I during AMI offers more accurate risk stratification of mortality risk than troponin alone.

## 1. Introduction

Troponin is currently considered the gold-standard biomarker for diagnosing and evaluating the extent of cardiac injury in acute myocardial infarction (AMI) patients [1,2,3,4,5,6,7]. Acutely after AMI, the systemic endocrine response adapts heart and global physiologic functions [8,9,10] and later directs cardiac remodeling [11,12]. Integrative physiologic analysis taking into account indicators of local myocardial injury and systemic response may offer more precise clinical evaluation.

Among numerous hormones, serum cortisol has been greatly investigated in AMI population [9,13,14,15,16]. Previous studies have shown that neuroendocrinal systems get activated at the early stage of AMI, sustaining homeostasis in the cardiovascular system. However, extended release of the cortisol hormone may lead to a detrimental effect [8,13]. Reports have been inconsistent, with different serum cortisol values being associated with contrasting outcomes [17,18]. Previous publications revealed that serum cortisol is dissociated from its anticipated tissue-level activity in patients diagnosed with critical illness [19,20]. A downstream approximation of cortisol activity may resolve the inconsistencies observed from the measurement of serum cortisol. The theory of Endobiogeny is an integrative physiologic approach to medicine [9,12]. On the grounds of this theory, researchers claim to have developed a method to evaluate cortisol activity by using elements of complete blood count with its differential (CBCD) [9,21]. This study aims to compare the prognostic value of the cortisol index of Endobiogeny with serum cortisol in patients diagnosed with AMI. Moreover, we seek to develop a more precise mortality prediction model by using both the cortisol index and troponin I levels in AMI patients.

## 2. Materials and Methods

### 2.1. Study Population

This prospective study included 123 consecutive patients diagnosed with AMI admitted to the intensive care unit of the Hospital of Lithuanian University of Health Sciences from March 2018 to May 2020. Diagnostic coronary angiography and revascularization was performed for all patients.

Acute myocardial infarction (AMI) was defined by the presence of symptoms consistent with cardiac ischemia and troponin I levels above the 99th percentile. Acute ST-elevation myocardial infarction (STEMI) was defined by the presence of criteria for MI plus one of the following: persistent ST segment elevation of >1 mm in two contiguous electrocardiographic leads or the presence of a new or a presumably new left bundle branch block. Non-ST-segment elevation myocardial infarction (NSTEMI) was defined by the presence of criteria for MI without ST elevation in the electrocardiogram.

Patients with the following before admission were excluded: previous history of acute coronary syndrome, prior revascularization, current utilization of systemic and inhalatory glucocorticoids, hypo- or hyperthyroidism, or any intervention (mechanical ventilation, central venous line, interventional angiography).

Written informed consent was obtained from all patients before the inclusion. The study protocol conforms to the ethical guidelines of the 1975 Declaration of Helsinki as reflected in prior approval by the Regional Biomedical Research Ethics Committee of the Lithuanian University of Health Sciences (ID No. BE-2-4, 29 March 2018).

### 2.2. Blood Samples

Venous blood samples were drawn by the standard venipuncture procedure for each study patient on admission (before any intervention), <36 h prior to discharge, and at six-month follow up evaluation. The samples were allowed to clot at room temperature, and sera were separated for analysis. The cortisol index was calculated by entering elements from the CBCD from admission, discharge and follow up by using proprietary software on Microsoft Excel (v16.16.7, Microsoft Corp, Redmond, Washington, DC, USA). The normal range of the cortisol index for adults is 3–7. For cortisol index calculation, we used T1 = admission, T2 = discharge, and T3 = 6-month follow up. ∆Cortisol index _1–2_ was calculated by subtracting the discharge cortisol index from the admission cortisol index. ∆Cortisol index _1–3_ was calculated by subtracting the 6-month follow-up cortisol index from the admission cortisol index. The serum concentrations of cortisol and troponin I were quantified by using commercial kits (ST AIA-PACK CORT, ST AIA-PACK cTnI 3rd-Gen, respectively) by automated enzyme immunoassay analyzer *AIA-2000* (Tosoh corporation, Tokyo, Japan) following the manufacturer’s recommendations. The cortisol normal value is 177–578 nmol/L at 7 a.m. and <434 nmol/L at 4 p.m. The cortisol index was calculated directly on admission, before discharge, and after 6 months. Troponin was measured directly on admission and 24 h after admission.

### 2.3. Follow-Up

After a 2-year period, all participants were contacted by phone to update information about their vital status and cardiovascular history.

### 2.4. Statistical Analysis

The continuous variables were expressed as the median (interquartile range) or the mean and standard deviation (M (SD)). The categorical variables were described as a percentage (number). The Mann-Whitney test was used to compare the quantitative sizes of two independent samples. The correlations were computed by using Spearman’s or Pearson’s method. Kruskal-Wallis test was used to compare non-normally distributed variables. The method of the receiver Operating Characteristics (ROC) curve was used. The ROC curves were assessed for the predicting outcomes. Survival curves were established by the Kaplan-Meier estimation method. The relative risk of death was estimated by using the Cox regression statistical model. The results were considered statistically significant when the two-tailed *p*-value was <0.05.

All the statistical analyses were performed with *SPSS 23.0* software (SDSPSS, Chicago, IL, USA).

## 3. Results

### 3.1. Baseline Characteristics

From March 2018 to May 2020, one hundred twenty-three patients diagnosed with AMI were enrolled. Of those, 64% had STEMI and 43% had anterior wall AMI. Mean age was 63.9 (SD 11.6 years), 65% were men. Onset to balloon time Median (25–75%) was 6.0 (3.0–14.0) hours. 46% of patients involved in the study were not taking any medication, while about 20% of the participants were on Angiotensin-converting enzyme inhibitors and Beta blockers. Other relevant basic characteristics of the population are presented in Table 1.

The glomerular filtration rate on admission mean was 73 (SD ± 20), whereas the Median (25–75) was 75 (61–89). The glucose on admission Median (25–75) was 6.7 (5.5–8.6). 40.7% of the patients had normal blood glucose levels on admission.

### 3.2. Onset of Pain to Admission Time

The highest mean of serum cortisol M (SD) 916.0 (295.3) (median (25–75%) 895.1 (809.1–1002.0) nmol/L was in those patients whose onset of pain to admission time was ≤3 h (Table 2).

The serum values declined with the longer onset of pain to admission time. In contrast, the cortisol index peak admission value was in the 7–12 h group, whereas the lowest value was in the ≤3 h group.

### 3.3. Serum Cortisol on Admission and Cortisol Index Variation during and after AMI

There was a positive moderate correlation between the cortisol index on admission with the cortisol index on discharge (*r* = 0.401; *p* = 0.001), and a weak correlation between the cortisol index on admission and the cortisol index after 6 months was observed (*r* = 0.275; *p* = 0.018) as assessed by the nonparametric Spearman’s rank correlation. In Table 3, frequencies of serum cortisol, cortisol index on admission, before discharge, and after 6 months are listed.

### 3.4. Serum Cortisol and Cortisol Index on Admission Correlation with Some Inflammatory Markers

There was a positive weak correlation between serum cortisol and white blood cells on admission (*r* = 0.2; *p* = 0.03) and neutrophil on admission (*r* = 0.2; *p* = 0.02). However, serum cortisol did not correlate with C-reactive protein. Moreover, there was a positive weak correlation between the cortisol index and C-reactive protein on admission (*r* = 0.2; *p* = 0.02).

### 3.5. Correlation between Troponin I with Serum Cortisol, Cortisol Index and ∆Cortisol Index

Serum cortisol inversely correlated with troponin I on admission (*r* = −0.410; *p* = 0.001), but directly correlated with troponin I peak (*r* = 0.376; *p* = 0.001). However, no significant correlation was found between the cortisol index on admission and troponin I on admission (*r* = −0.007; *p* = 0.935), or between cortisol index on admission and troponin I peak (*r* = 0.163; *p* = 0.073) (Table 4).

### 3.6. Serum Cortisol, Cortisol Index and Troponin I Results and Mortality

In Table 5, comparative analysis between serum cortisol and cortisol index on admission (Mann-Whitney test) is presented. When comparing survivors and non-survivors, serum cortisol values were higher in survivors, whereas the cortisol index was higher in non-survivors.

The median values of 25–75% are given. The *p*-value is based on the non-parametric Mann-Whitney test.

Based on the Receiving Operating Characteristics (ROC) test (Table 6), the cut-off mortality value for serum cortisol was <800 nmol/L (with sensitivity of 100% and specificity of 51.2%) (Figure 1), while the cut-off mortality value for cortisol index was >100 (with sensitivity of 83.3% and specificity of 52.7%) (Figure 2) and troponin I on admission value > 1.56 μmol/L (with sensitivity of 81.8% and specificity of 56.8%) (Figure 3).

According to Kaplan-Meier survival curve, the survival of the patients with cortisol index < 100 was 98.3 ± 1.7, whereas the survival of the patients with cortisol index > 100 was 82.5 ± 4.8 (Figure 4).

The relative risk (RR) of mortality assessed by cortisol index was 8.916 (1.082–73.493) (when the cortisol index by Cox prognostic analysis was adjusted to age and sex).

Admission troponin I > 1.56µg/L showed a relative risk (RR) of death of 5.326 (1.15–24.655) (Figure 5).

### 3.7. RR of Mortality in Serum Cortisol with Troponin I Group and Cortisol Index with Troponin I Group

All the patients survived when troponin I on admission was <1.56 μmol/L and serum cortisol on admission was >800 nmol/L, or else when troponin was I > 1.56 μmol/L and serum cortisol was >800 nmol/L. However, 2 (11.8%) patients died with troponin I < 1.56 μmol/L and serum cortisol < 800 nmol/L, and 7 (21.2%) patients died when troponin I on admission was >1.56 μmol/L and serum cortisol on admission was <800 nmol/L (χ^2^ = 9.715, df = 3, *p* = 0.02, Chi square, Monte Carlo). OR (95% CI) was higher 7.808 (1.517–40.175) when troponin I on admission was >1.56 μmol/L and serum cortisol on admission was <800 nmol/L.

The patients’ results were divided into 4 groups according to the cortisol index and troponin I mortality cut-off value on admission. All the patients survived when troponin I on admission was <1.56 μmol/L and the cortisol index on admission was <100. However, 2 (9.1%) patients died with troponin I > 1.56 μmol/L and cortisol index < 100. Similarly, 2 (7.7%) patients died when troponin I on admission was <1.56 μmol/L and the cortisol index on admission was >100. Finally, 7 (20%) patients died when troponin I measured >1.56 μmol/L and the cortisol index showed >100 (χ^2^ = 8.902, df = 3, *p* = 0.031) (Table 7, Figure 4 and Figure 5). According to Cox regression analysis, troponin > 1.56 μmol/L and cortisol index > 100 remained associated with the 2-year mortality (RR: 4.6, 95% CI: 1.345–15.726, *p* = 0.015).

## 4. Discussion

We performed a prospective single center study to evaluate two hypotheses. The first was that the cortisol index would more accurately predict 24-month mortality than the admission serum cortisol. The second hypothesis was that an integrated physiologic approach classifying patients by local myocardial injury and systemic endocrine response would more accurately stratify the risk of mortality. Our results confirmed both hypotheses thus suggesting that the cortisol index derived from the theory of Endobiogeny may be a helpful diagnostic and prognostic tool for patients post-AMI.

Our study is the first to evaluate the cortisol index after myocardial infarction. We found the peak cortisol index at 7–12 h after the onset of AMI pain. The time to peak serum cortisol in literature is inconsistent. In our study, serum cortisol peaked <3 h after the onset of AMI pain. De la Hoz et al. reported 8 h [22], while Vetter et al. reported 30–60 min from the onset of symptoms [14]. The peak serum cortisol is quantitative evaluation of the adrenal gland output; it lacks linear correlation with the cortisol function in critical illness [19,20,23]. The peak cortisol index evaluates the maximum tissue-level *activity*. Cortisol has strong affinity for cortisol and mineralocorticoid receptors (MR) [24]. When elevated, cortisol occupies MR in preference to aldosterone [25]. With insufficiency of the converting enzyme 11β-hydroxysteroid dehydrogenase (11βHSD), the effects of cortisol are amplified in injured tissues to a degree not anticipated by serum measurement [26]. Thus, we conjecture that the later time to peak cortisol index may reflect the net effects of the cortisol action on cortisol and mineralocorticoid receptors as well as the role of insufficient 11βHSD in the target organs. If this hypothesis is correct, a down-stream assessment of the cortisol activity, such as with the cortisol index, may be preferable to serum measurement.

The cortisol index was sensitive to acute and chronical physiologic evolution post-AMI. It was elevated on admission, lowered significantly at discharge, and further lowered at 6 months with the mean value being chronically elevated. Hedayat et al. produced similar findings when evaluating the cortisol index in patients with the heart failure of different etiologies and the New York Heart Association (NYHA) 2–3 functional classification [21]. Serum cortisol is not routinely measured post-AMI. As CBCD is routinely measured, the cortisol index can be calculated repeatedly and monitored in patients as a novel and inexpensive marker.

The onset of pain to admission time is the gold standard scale for estimating myocardial damage and patient outcomes [27,28,29,30]. In our study, admission serum cortisol moderately correlated with mortality based on this criterion. Other studies were once again inconsistent. The discrepancies may be due to the employed methodology. Many of these studies evaluate cortisol without considering the onset of pain to admission time [13,17,18,22]. The cortisol index also correlated with 24-month mortality. Future studies need to be performed with a larger sample size so that to determine if these results can be reproduced. As the cortisol index can be calculated post-hoc by using data from prior studies in which CBCD was measured on admission, a large-scale retrospective study can be easily conducted as well.

In this study, we established a cut-off value for the cortisol index of >100 in patients for the increased risk of mortality. While troponin levels have shown to be a useful tool for prognostic stratification of patients with MI, its main disadvantage is the prolonged period (3–9 hrs) till the peak troponin I release [31,32,33]. The cortisol index may be valuable at the discrimination of high versus low risk patients after AMI. The results of other studies demonstrate that other prognostic markers, such as troponin I and T, natriuretic brain peptides, cortisol, and even the genetic background, may be important if we wish to assess the risk of the fatal outcome [34]. However, our data demonstrates that the cortisol index if comparing to troponin I and cortisol is the most valuable predictor of mortality after AMI with high sensitivity and specificity. Our findings can aid cardiologists during daily practice to early (<3 h) stratification of the patients with AMI when using the cortisol index as a valuable prognostic marker.

With respect to the question of an integrative physiologic approach, we found that evaluating both the cortisol index and troponin I allowed for good risk stratification of mortality. Those with the worst responses (cortisol index > 100, troponin I > 1.56 μg/L) had the highest rate of 24-month mortality. Those with more favorable outcomes had lower risks of mortality, with no deaths reported in those with cortisol index < 100 and troponin I < 1.56 μg/L. Interestingly, we observed that if only cortisol or troponin were elevated, the risk of mortality was equivalent within 24 months, and intermediate compared to those with both indicators being high or low. This observation lends support to the value of the global systems approach to integrative physiology, in which, the sequelae of organ injury is contextualized to the systemic response to that injury as a determinant of survival and recovery.

## 5. Limitations

The study sample was small given the low mortality rate.

## 6. Conclusions

Our study produced two observations for the first time. Firstly, the cortisol index is more predictive of mortality than serum cortisol. Second, combined assessment of the cortisol index and troponin I, offers more accurate risk stratification of the mortality risk than troponin I alone. Because of the novelty of the application of the cortisol index to AMI, further research is recommended to confirm these results by employing a multi-center prospective clinical trial.

## Figures and Tables

**Figure 1 medicina-57-00602-f001:**
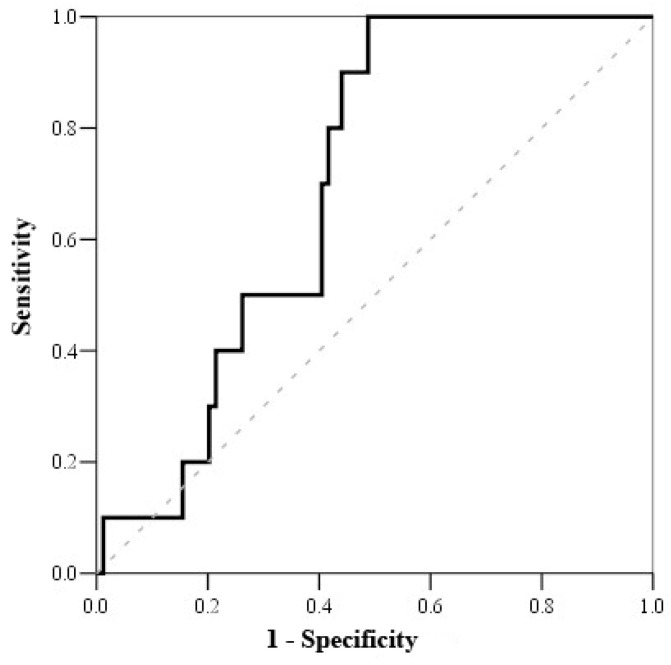
ROC curve of serum cortisol on admission.

**Figure 2 medicina-57-00602-f002:**
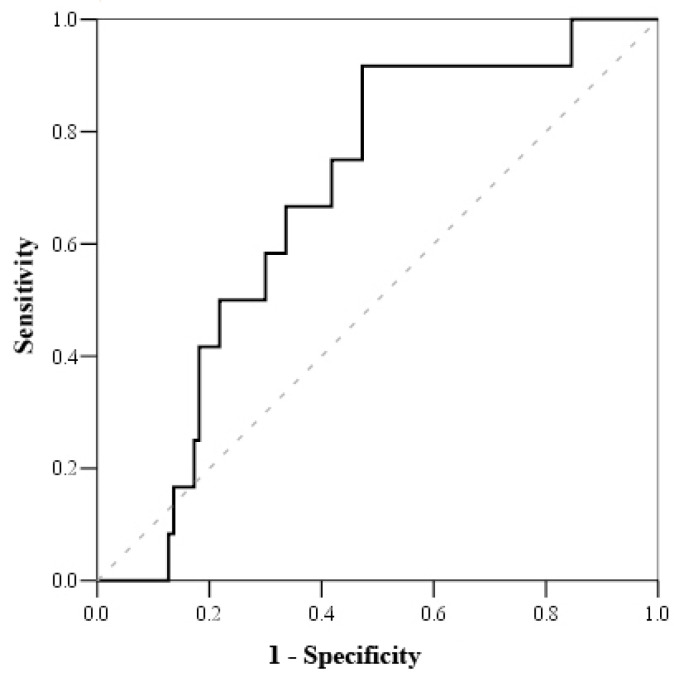
ROC curve of cortisol index.

**Figure 3 medicina-57-00602-f003:**
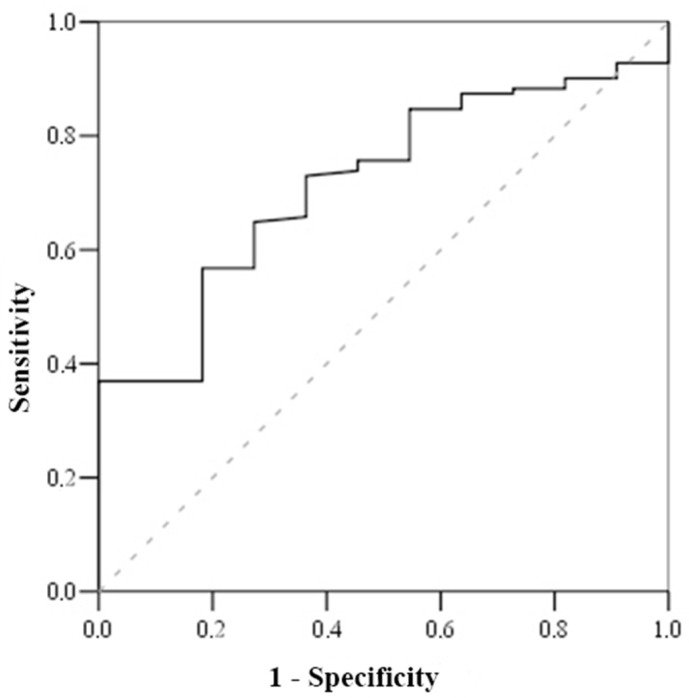
ROC curve of troponin.

**Figure 4 medicina-57-00602-f004:**
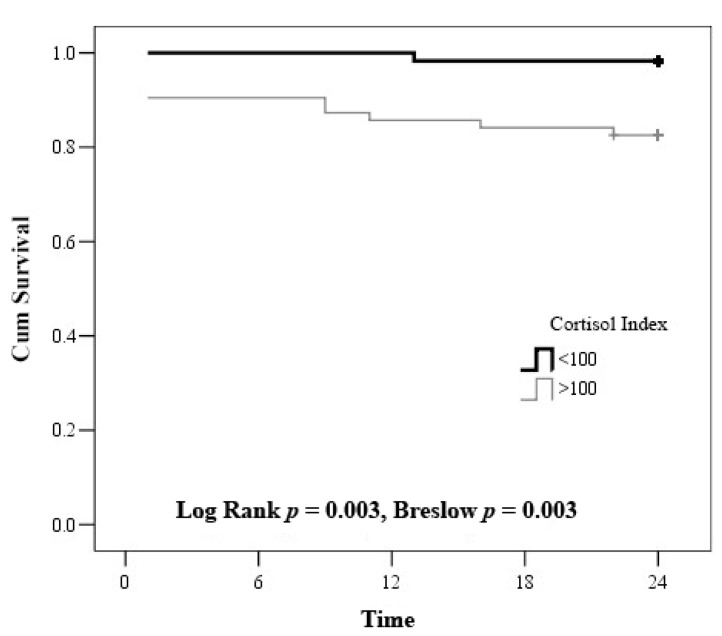
Kaplan–Meier curve—estimation of the time to death by cortisol index.

**Figure 5 medicina-57-00602-f005:**
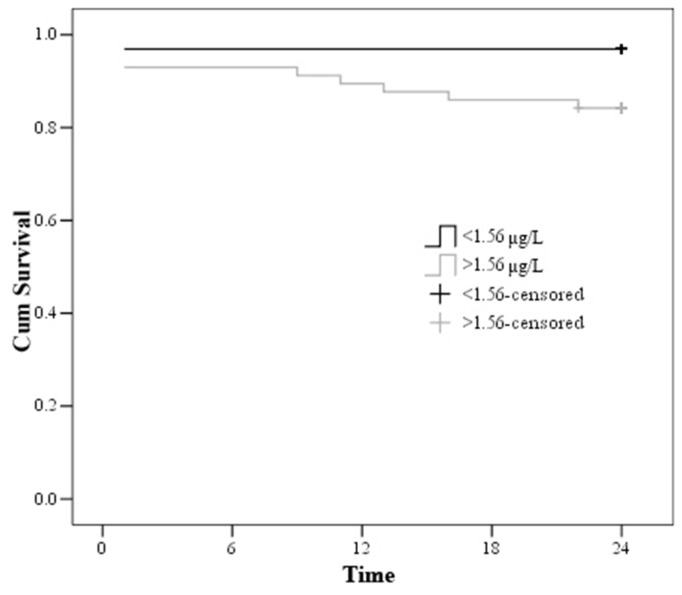
Kaplan–Meier curve—estimation of the time to death by troponin on admission.

**Table 1 medicina-57-00602-t001:** Baseline characteristics of study participants (*n* = 123).

Characteristics	Category	Number (Percentage)
Age, years M (SD)		63.9, (SD ± 11.6)
Gender, *n*, (%)	Male	80 (65)
	Female	43 (35)
Type of MI, *n*, (%)	STEMI	79 (64)
	Non STEMI	44 (36)
Type of MI, *n*, (%)	Anterior wall	53 (43)
	Posterior wall	48 (39)
	Lateral wall	22 (18)
Comorbidities, *n*, (%)	Diabetes mellitus	16 (13)
	Hypertension	112 (91)
	Dyslipidemia	78 (63)
	CKD *	5 (4)
	Old CVA	12 (10)
	COPD	5 (4)
Killip classes, *n*, (%)	I	48 (39)
	II	65 (53)
	III	4 (3)
	IV	6 (5)
Onset to balloon time (in hours), *n*, (%)	<3	41 (33)
	4–6	22 (18)
	7–12	21 (17)
	13–48	38 (31)
Ongoing pharmocological treatment, *n*, (%)	ACE inhibitors	25 (20)
	ARB	7 (6)
	Nitrates	7 (6)
	CCB	12 (10)
	Beta blockers	27 (22)
	Anticoagulant	3 (2)
	Antiplatelet drugs	7 (6)
	Statins	3 (2)
	Diuretics	9 (7)
	MRB	1 (1)

M-mean, MI-myocardial infarction, STEMI—ST elevation myocardial infarction; CVA—cerebrovascular accident; CKD—chronic kidney disease; COPD—chronic obstructive pulmonary disease; ACE—Angiotensin-converting enzyme; ARB—Angiotensin receptor blockers; CCB—Calcium channel blockers; MRB—Mineralocorticoid receptor blockers; CKD * was defined as an estimated glomerular filtration rate less than 30 mL/min per 1.73 m^2^ on admission.

**Table 2 medicina-57-00602-t002:** Mean values of serum cortisol, cortisol index and ∆cortisol index distributed according to onset of pain to admission time.

	Onset of Pain to Admission Time (in Hours)				*p* Value
	≤3	4–6	7–12	>13	
	Median (25–75)				
Serum Cortisol _1_ (nmol/L)	895.1 (809.1–1002.0) ^ab^	778.7 (566.7–1157.7) ^c^	539.8 (350.3–846.9) ^a^	505.7 (388.1–713.9) ^bc^	χ^2^ = 21.599; df = 3; *p* < 0.001
Cortisol index _1_	40.0 (14.0–131.2) ^ab^	190 (104.3–522.9) ^ac^	251.0 (93.0–504.7) ^bd^	76.6 (30.2–281.4) ^cd^	χ^2^ = 19.296; df = 3; *p* < 0.001
Cortisol index _2_	15.4 (8.5–26.0) ^a^	17.6 (13.9–43.6) ^a^	17.2 (7.3–31.4)	15.0 (7.3–38.1)	χ^2^ = 2.394; df = 3; *p* = 0.495
Cortisol index _3_	7.0 (4.7–14.6) ^a^	14.8 (8.1–21.3) ^a^	10.7 (7.3–15.7)	12.2 (5.8–24.1)	χ^2^ = 4.833; df = 3; *p* = 0.184
∆Cortisol index _1–2_	22.8 (1.4–89.3) ^ab^	176.7 (99.8–346.7) ^ac^	198.9 (58.5–485.6) ^bd^	39.3 (10.8–139.9) ^cd^	χ^2^ = 17.3; df = 3; *p* = 0.001
∆Cortisol index _1–3_	27.5 (5.4–140.0) ^a^	218.0 (97.5–530.1) ^ac^	159.3 (55.4–492.9)	36.4 (7.3–169.7) ^c^	χ^2^ = 10.44; df = 3; *p* = 0.015

_1_—on admission, _2_—before discharge, _3_—after 6 months; ^a–d^*p* < 0.05.

**Table 3 medicina-57-00602-t003:** Frequencies of serum cortisol, cortisol index on admission, before discharge, and after 6 months.

	Serum Cortisol _1_ (nmol/L)	Cortisol Index _1_	Cortisol Index _2_	Cortisol Index _3_
*N*	94	122	83	75
M (SD)	753.1 (351.8)	242.5 (342)	26.4 (32)	16.3 (20)
Median (25–75%)	778.5 (466.6–955.4)	102.6 (38.7–291.9) ^ab^	15.4 (8.7–31.9) ^ac^	10.7 (5.9–19.3) ^bc^

_1_—on admission, _2_—before discharge, _3_—after 6-month period; ^a–c^*p* ≤ 0.001.

**Table 4 medicina-57-00602-t004:** Correlation between troponin I on admission and troponin I peak with serum cortisol, cortisol index on admission and ∆cortisol index _1–2_ assessed by nonparametric Spearman’s rank correlation.

	Troponin I _1_	Troponin I Peak
Serum cortisol _1_	*r* = −0.410; *p* = 0.001	*r* = 0.376; *p* = 0.001
Cortisol index _1_	*r* = −0.007; *p* = 0.935	*r* = 0.163; *p* = 0.073
∆Cortisol index _1–2_	*r* = 0.358; *p* = 0.001	*r* = 0.247; *p* = 0.023
∆Cortisol index _1–3_	*r* = 0.195; *p* = 0.097	*r* = 0.233; *p* = 0.046

_1_—on admission, _2_—before discharge, _3_—after 6 months.

**Table 5 medicina-57-00602-t005:** Differences between serum cortisol, cortisol index on admission and Troponin I on admission (Mann-Whitney test) in the survivors and non-survivors groups.

Parameters	Survivors Group *n* = 111/24	Non-Survivors Group *n* = 12	*p* Value
	Median (25–75%)		
Serum cortisol _1_	817.7 (476.3–978.2) 887.3 (558.6–971.5) *	595.4 (416–722.2)	0.039 0.02 *
Cortisol index _1_	89 (35.5–261.6) 75.9 (13.7–276) *	253.0 (104.2–454.2)	0.043 0.048 *
Troponin _1_	1.08 (0.1–3.96) 1.03 (0.07–2.67) *	4.36 (1.58–11.53)	0.018 0.012 *

* *n* = 24 subjects were selected randomly from the survivor group (*n* = 111) to increase the reliability of the study; _1_—on admission.

**Table 6 medicina-57-00602-t006:** Prognostic values and its characteristics distributions of serum cortisol, cortisol index and troponin I ROC test according to mortality.

Parameters/ Optimal Meaning of Its Change	AUC-ROC Curve (%)	Sensitivity/Specificity	Henly-McNeil Method z Score *p* Value	Survival/Death (%)	*p* Value	Death RR (95% CI)
Serum cortisol _1_ <800 nmol/L	70.0	100 51.2	3.27 0.001	0 19.6	0.002	-
Cortisol index _1_ >100	78.0	83.3 52.7	2.7 0.007	3.3 15.8	0.018	5.179 (1.134–23.65)
Troponin I _1_ > 1.56 μg/L	71.7	81.8 56.8	3.3 0.001	3.1 15.8	0.014	5.326 (1.15–24.655)

_1_—on admission, RR—relative risk, CI—confidence interval, AUC—ROC curve—area under the Receiver Operating Characteristics curve.

**Table 7 medicina-57-00602-t007:** Mortality in cortisol index with troponin I group.

	Troponin I > 1.56 μmol/L and Cortisol Index > 100 μmol/L (*n* = 35)	Troponin I < 1.56 μmol/L and Cortisol Index > 100 (*n* = 26)	Troponin I > 1.56 μmol/L and Cortisol Index < 100 (*n* = 22)	Troponin < 1.56 μmol/L and Cortisol Index < 100 (*n* = 38)
	*n* (%)			
Non-survivors group	7(20.0) *	2(7.7)	2(9.1)	0
		4(4.7) *		

χ^2^ = 8902, df = 3, *p* = 0.031, * *p* = 0.008, OR (95% CI)—5.125 (1.395–18.828).

## Data Availability

The datasets generated for this study are available on request to the corresponding author.

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
