# Peer review of "Prognostic Value of Cortisol Index of Endobiogeny in Acute Myocardial Infarction Patients"

_medicina, 2021, doi:10.3390/medicina57060602_

Round 1

Reviewer 1 Report

  • Sample size analysis should be reported in statistical analysis. Otherwise, this should be indicated as a pilot study.
  • Table 1: authors should report here ongoing pharmacological treatments.
  • Table 1: Do authors have glomerular filtration rate of the cohort? It is surprising that CKD was not so prevalent (at least prevalence below the expected): how was CKD defined?
  • Do authors have some immune-inflammatory marker to be correlated with changes in cortisol over the observational period? Please, comment on the possible relevance of very acute phase proteins (e.g. C-reactive protein of hepatic origin, neutrophils-derived Pentraxin 3 or some acute phase protein from the kidney or from the surrenal gland).
  • Do authors have further information about glucose and other biochemical features? This would better depict the clinical milieu of the investigation.
  • Figure 4-6: Kaplan-Meier each curves should start from the same y-axis.
  • Figure 1-3: ROC curves should be compared for their efficacy: a) vs a basal model without the inclusion of the tested marker and b) vs each other. Please use appropriate analysis (DeLong method or the Hanley-McNeil one (Radiology 1986) to compare the areas-under the curve. Without this, it is very hard to support conclusions from these data. In addition, youden c-statistic to derive the best threshold value of the marker to maximally identifies the outcome should be provided. This aspect would be more properly of clinical value. This part of results and discussion has to be deeply revised.
  • Delta-values of cortisol. Statistically, authors state that each delta was gathered by subtracting the last vs the first cortisol determination. I suggest to revise these ratios by dividng this difference for the individual basal value of cortisol. This should better normalize the rate of change among individuals. If this has been still done, please better specify this because it is not clear in the text.
  • Was the observational period exactly the same for all individuals? If yes, please clarify better in the text the rationale of using the Cox regression model.
  • Page 3/12, lines 86-88: what do authors refer for “normal” cortisol values?

Author Response

  1. Sample size analysis should be reported in statistical analysis. Otherwise, this should be indicated as a pilot study.
    • Sample size is already reported in the materials and method paragraph (2.1 study population);

  1. Table 1: authors should report here ongoing pharmacological treatments.
    • According to the reviewer comment we added the ongoing medications to the manuscript (tabale 1)
  1. Table 1: Do authors have glomerular filtration rate of the cohort? It is surprising that CKD was not so prevalent (at least prevalence below the expected): how was CKD defined?
    • We redefined the CKD as GFR < 30 ml/min/per1.73 m2 , and accordingly we corrected the data presented in tabale 1
  1. Do authors have some immune-inflammatory marker to be correlated with changes in cortisol over the observational period? Please, comment on the possible relevance of very acute phase proteins (e.g. C-reactive protein of hepatic origin, neutrophils-derived Pentraxin 3 or some acute phase protein from the kidney or from the surrenal gland).
    • We added section 3.4, to discuss the correlation we have established between serum cortisol, cortisol index and immune-inflammatory markers
  1. Do authors have further information about glucose and other biochemical features? This would better depict the clinical milieu of the investigation.
    • Data regarding glucose levels were added in section 3.1
  1. Figure 4-6: Kaplan-Meier each curves should start from the same y-axis.
    • Figures were corrected according to the reviewer request
  1. Figure 1-3: ROC curves should be compared for their efficacy: a) vs a basal model without the inclusion of the tested marker and b) vs each other. Please use appropriate analysis (DeLong method or the Hanley-McNeil one (Radiology 1986) to compare the areas-under the curve. Without this, it is very hard to support conclusions from these data. In addition, youden c-statistic to derive the best threshold value of the marker to maximally identifies the outcome should be provided. This aspect would be more properly of clinical value. This part of results and discussion has to be deeply revised.
    • We have added statistical analysis according to Hanley- McNeil method into tabale 6.  
  1. Delta-values of cortisol. Statistically, authors state that each delta was gathered by subtracting the last vs the first cortisol determination. I suggest to revise these ratios by dividng this difference for the individual basal value of cortisol. This should better normalize the rate of change among individuals. If this has been still done, please better specify this because it is not clear in the text.
    • We have now clarified this matter more pricisely, and accordingly new corrections was made in section 2.2
  1. Was the observational period exactly the same for all individuals? If yes, please clarify better in the text the rationale of using the Cox regression model.
    • The follow up period was exactly the same for all participants in the study ( 2 years), cox regression model was used to show the outcomes of patient according to cortisol index and troponin levels.
  1. Page 3/12, lines 86-88: what do authors refer for “normal” cortisol values?
    • The normal value of cortisol we refered in the text, was the level set by the labarotary kit used to test serum cortisol.

Reviewer 2 Report

Dear authors,

There is a very interesting topic debated in your article, and it was less debated in the medical literature.

Thus, there are several aspects that are requiring clarifications.

  • Row 42: you could explain more over the role and fluctuations of cortisol during acute myocardial infarction and why is it relevant for the prognosis
  • Row 67, also inhalatory corticoids, or only systemic ones? What about patients with Cushing disease? How long before should the patient not have any intervention?
  • Row 75 – which hour were the samples drown, as recommended in the morning?
  • Row 137 – it is interesting why the 2 laboratory results did not correlate
  • Conclusions should be shorter and reflect a summary of your findings

Please revise English, there are several punctuation and spelling issues, and also some phrases are strangely formulated.

Author Response

  1. Row 42: you could explain more over the role and fluctuations of cortisol during acute myocardial infarction and why is it relevant for the prognosis
    • We have added additional information within the text discussing the role of fluctuation of cortisol during AMI
  1. Row 67, also inhalatory corticoids, or only systemic ones? What about patients with Cushing disease? How long before should the patient not have any intervention?
    • Patients who were in inhilatory corticoids and who had a hormonal disease ( cushing disease) or any intervention were excluded from our study, according to reviewer comment we have emphasized this point within the manuscript.
  1. Row 75 – which hour were the samples drown, as recommended in the morning?
    • The blood was drawn as soon as the patient was admitted to the ICU unit before undergoing PCI. Thereby no specific fixed time was assigned for blood withdrawal
  1. Row 137 – it is interesting why the 2 laboratory results did not correlate
    • We have changed the text to make the result more clear for the reviewer and the reader.
  1. Conclusions should be shorter and reflect a summary of your findings
    • We have shortened the conclusion according to the reviwer request.
